# Time-Reversal Provides Unsupervised Feedback to LLMs

**Varun Yerram** * † §
Google DeepMind

**Rahul Madhavan** * ‡
Indian Institute of Science

**Sravanti Addepalli** * † §
Google DeepMind

**Arun Suggala** †
Google DeepMind

**Karthikeyan Shanmugam** † §
Google DeepMind

**Prateek Jain** †
Google DeepMind

## Abstract

Large Language Models (LLMs) are typically trained to predict in the forward direction of time. However, recent works have shown that prompting these models to look back and critique their own generations can produce useful feedback. Motivated by this, we explore the question of whether LLMs can be empowered to think (predict and score) backwards to provide unsupervised feedback that complements forward LLMs. Towards this, we introduce Time Reversed Language Models (TRLMs), which can score and generate queries when conditioned on responses, effectively functioning in the reverse direction of time. Further, to effectively infer in the response to query direction, we pre-train and fine-tune a language model (TRLM-Ba) in the reverse token order from scratch. We show empirically (and theoretically in a stylized setting) that time-reversed models can indeed complement forward model predictions when used to score the query given response for re-ranking multiple forward generations. We obtain up to $5\%$ improvement on the widely used `AlpacaEval` Leaderboard over the competent baseline of best-of-N re-ranking using self log-perplexity scores. We further show that `TRLM` scoring outperforms conventional forward scoring of response given query, resulting in significant gains in applications such as citation generation and passage retrieval. We next leverage the generative ability of `TRLM` to *augment* or provide unsupervised feedback to input safety filters of LLMs, demonstrating a drastic reduction in false negative rate with negligible impact on false positive rates against several attacks published on the popular JailbreakBench leaderboard.

## 1   Introduction

Large Language Models (LLMs) trained on a large corpora of text are able to accomplish a wide variety of downstream tasks such as summarization, open-ended/ context-based question answering, document retrieval, and citation generation [Brown et al., 2020, Zhao et al., 2023a]. While the generations from pre-trained and instruction-tuned models already show significant promise, alignment techniques such as Reinforcement Learning via Human Feedback (RLHF) [Anil et al., 2023a, Ouyang et al., 2022] are widely used to improve the quality of their generations further. However, these methods rely heavily on additional supervision to construct preference data, which can be expensive to acquire, or noisy for training. This brings up a natural question – *Can we generate useful feedback on LLM generations without additional supervised data?*

---

*Equal Contribution.
†Work done as part of Google Research
‡Work done as a Student Researcher at Google Research
§Correspondence to: vyerram@google.com, sravantia@google.com, karthikeyanvs@google.com

38th Conference on Neural Information Processing Systems (NeurIPS 2024).

A recent line of work aims at *specially prompting* LLMs to review their own generations and generate meaningful natural language feedback, which can subsequently be used to refine them [Madaan et al., 2024]. This process can be repeated to improve the generations iteratively. The success of such methods serves as an evidence that it is indeed possible to obtain better responses without additional supervision. However, such methods rely on the superior instruction following and reasoning abilities of LLMs, which may not necessarily hold for low capacity models. Further, these methods involve sequential processing of the generated responses, and thus increase inference time significantly.

In this work, we propose a natural method of enabling LLMs to *look backwards* in order to obtain meaningful unsupervised feedback during inference. Towards this, we introduce a class of models that we call *Time Reversed Language Models* (TRLMs), which operate in the reversed direction of a regular LLM, or the *time-reversed* direction. Rather than predicting (or scoring) in the standard `query` → `response` direction, time reversed language models predict (or score) in the `response` → `query` direction. We first introduce `TRLM-Fo` - a TRLM variant based on forward models, which are *prompted* to operate in the time-reversed direction using a prompt such as `"Generate a question that would result in the following answer:  <response>"`. Further, we extend the reversal to *token*-level granularity by pre-training LLMs from scratch in a reversed token direction, rather than the standard forward token direction. We call this as `TRLM-Ba` where `Ba` stands for Backward. Note that the inputs and outputs of such a model are in the reversed language order. Pre-training `TRLM-Ba` on reversed text exposes the model to a completely different world model where the conventional order of information is flipped. Introductions *follow* conclusions, questions *follow* answers, logical precedents *follow* their antecedents. Hence, such a model may not only develop representations that are distinct from those of a regular LLM – despite being trained on the same pre-training corpus – but may also be better suited to score/ generate in the reverse direction, i.e. conditional on the response.

We show in several use-cases that scoring and generation in this reverse direction can produce non-trivial feedback on the responses generated by forward LLMs. We consider three classes of tasks to showcase the scoring and generating capability of TRLM, viz. a) Reranking answers in open ended question answering b) Citation and retrieval tasks and c) Amplifying existing safety filters through query generation in the reverse.

**Our Contributions:**
**a)** We propose time reverse language models - `TRLM-Fo`, `TRLM-Ba` and `TRLM-FoBa`, all of which score and generate queries given responses, enabling their use in obtaining unsupervised feedback on LLM generations. `TRLM-Fo` is a forward model prompted to predict in reverse, while `TRLM-Ba` is pre-trained in the reverse token order, enabling reverse prediction naturally. `TRLM-FoBa` is pre-trained in both reverse and forward token orders and can be used to predict in forward or reversed language.
**b)** We demonstrate significant improvements when best-of-N reranking is applied to multiple LLM generations by using TRLM scores. Specifically, we show up to a $5\%$ improvement over self-reranking using `TRLM-Ba`, in LC win-rates (0.98 Pearson correlation with human preferences) against a `GPT4-1106-Preview` reference model. We show multiple ablations on this study.
**c)** We demonstrate that the reverse direction of scoring (`response` → `query`) is highly significant, as it improves citation attribution accuracy by 44.15% when compared to the forward baseline on the CNN-Daily Mail dataset. Further, we improve the NDCG@10 metric by $44.13$ points on the NF-Corpus medical information retrieval benchmark, and obtain similar improvements on MS-Marco as well.
**d)** We show that the reverse generation capability of the `TRLM` models - specifically `TRLM-Ba`, can be used to improve False Negative rate (FNR) of input safety filters with negligable impact on FPR. We show significant improvements on several attacks submitted to the Jailbreakbench benchmark, and on a Human Annotated dataset from JailbreakBench.

We complement these results with theoretical arguments using a bipartite graph model between queries and responses, to show that RLHF done with `TRLM-Ba` scores induces a non trivial distribution shift in answers, mitigating primitive forms of "hallucination" under the defined conditions.

## 2   Related Work

**Reverse Direction in Language Modeling:** Classical work [Serdyuk et al., 2017] showed how sequence to sequence models can regularize the current word token embedding based on the ability of the future tokens to be able to predict the current token. Such bi-directional (forward and reverse) consistency checks have been used to improve forward models. Golovneva et al. [2024] train an LLM

in the forward direction first, followed by the reverse token direction, and show that this alleviates the reversal curse identified by Berglund et al. [2023]. This work is closely related to ours in that we also consider a variant of combining reverse and forward token order during training. Our key models differ from this, and are trained in either forward (`TRLM-Fo`)/ reverse (`TRLM-Ba`) token order, using which we demonstrate improvements in a wide range of applications such as long form question answering, citations, retrieval and augmenting input filters for defending against toxic questions. Yang et al. [2023] use question generation from a given answer combined with access to external databases to determine hallucination. Another recent work [Guo et al., 2024] also explores a different pre-training order. While their focus is to correct causal ordering bias, our work instead is focused on the value that scoring and generation of these models bring to downstream tasks.

**Reversed scoring:** Several prior works [Li et al., 2016, Zhang et al., 2018, 2020] have proposed to improve the diversity of generated responses by optimizing the *mutual information* between the responses and the respective queries. These works motivate the need for better decoding strategies based on scores in both, `response` → query and `query` → `response` directions. We theoretically show that reverse scoring alone, when used with forward generations, will achieve this naturally using a formal RLHF based argument (Lemma 2), and present strong empirical results across a wide range of tasks to support the same.

**Controlling Decoding through feedback:** A broad line of works align a pre-trained model to a reward model trained on human feedback by using Reinforcement learning (RL) techniques like Proximal Policy Optimization (PPO) [Stiennon et al., 2020, Ouyang et al., 2022, Korbak et al., 2022], (Identity policy optimization) IPO (and $\Psi$PO) [Azar et al., 2024], Direct Preference Optimization [Rafailov et al., 2024] and offline RL [Snell et al., 2022]. Zhao et al. [2022] and Zhao et al. [2023b] calibrate likelihood of generated responses on a dataset with desired responses or human preference feedback.[Krause et al., 2020, Yang and Klein, 2021, Qin et al., 2022] control the generation of an LLM at test time by specifying constraint functions or discriminators that operate in the token or logit space, encouraging certain attributes in the output. Using preference feedback, Mudgal et al. [2023b] train a prefix scorer model that acts as a value function over partial completions consistent with the preference rewards. Yang et al. [2024b] investigate the relation between best-of-N-reranking and KL regularized RL objective. An observation made by Yang et al. [2024b] is that best-of-N-reranking dominates/ competes very well with most RL based alignment methods. Under certain assumptions, authors show formally that best-of-N-reranking approximates the optimal solution to the regularized RL objective. We take inspiration from this and use best-of-N-reranking to evaluate generations through unsupervised feedback by the reverse LLMs. Our work differs from all these in that they rely on external feedback to control generation, while our method does not.

**Self Play and Self Tuning:** Chen et al. [2023] explore how an LLM can be prompted to self-debug based on an explanation of the code produced by the LLM during code generation and the execution output on test cases. Welleck et al. [2022] use a corrector model that is trained to prefer a new corrected answer if the corrected answer has higher value that a default generation. They require access to a value function for this determination. All these approaches use an external feedback to align the model in their pipeline.

Fu et al. [2023] explore LLM agents initialized as buyers and sellers to play a negotiating game of setting the price of a transaction. A critic LLM provides feedback to both the buyer and seller agents to improve. Madaan et al. [2024] propose a self refining loop where the same model is prompted to provide feedback and further use the feedback to refine and regenerate. Both these works use very powerful and large models from the Claude, GPT-4, GPT-3.5 family to use self generated language feedback. Madaan et al. [2024] remark that the self refining approach does not work well with weaker models. In contrast, we focus on improving generation quality of much smaller models using unsupervised scalar feedback. Other prior works relating to self play are reviewed in the survey article by Amini et al. [2022].

## 3 TRLM - Time Reversed Language Models

We introduce our primary contribution - `TRLM` (**T**ime **R**eversed **L**anguage **M**odels), a class of language models that operate in the `response` → `query` direction during scoring and generation. This is achieved by either (a) [`TRLM-Ba`] reversing the token order and effectively utilizing previous token prediction instead of next token prediction during pre-training, scoring, and generation, or (b)

Table 1: Description of different TRLM model variants.

| Model | Description |
|---|---|
| TRLM-Ba | Pre-trained in the reverse token order for previous token prediction (Alg. 1 in the supplement). Instruction-tuned variant is FLaN fine-tuned [Longpre et al., 2023] in reverse token order. Scores the reversed question given a reversed answer combined with suitable prompts. Generates questions in the reverse direction when conditioned on answers in the reverse direction.

**Scoring:** $\mathbb{P}_{\texttt{TRLM-Ba}}\big(\text{Reverse}(\texttt{Scoring Prompt}+\text{Query}) \mid \text{Reverse}(\texttt{Conditioning Prompt} + \text{Answer})\big)$ (Alg. 2 in the supplement).
**Generation:** $\mathbb{P}_{\texttt{TRLM-Ba}}\big( \cdot \mid \text{Reverse}(\texttt{Conditioning Prompt} + \text{Answer})\big)$ |
| TRLM-Fo | Pre-trained in the usual forward token order. Scores `Question` given `Answer` using the prompt. Generates from the conditional distribution of an answer.

**Scoring:** $\mathbb{P}_{\texttt{TRLM-Fo}}\big(\text{Query} \mid \text{Answer} + \texttt{Conditioning Prompt} \big)$ (Alg. 3 in the supplement)
**Generation:** $\mathbb{P}_{\texttt{TRLM-Fo}}\big( \cdot \mid \text{Answer} + \texttt{Conditioning Prompt}\big)$ |
| TRLM-FoBa (Reverse) | Pre-trained both in forward and reverse token order (Alg. 4 in the supplement). Understands text in both directions. Reverse version scores and generates identically to `TRLM-Ba`.

**Scoring:** Scores identically to `TRLM-Ba`.
**Generation:** Generates identically to `TRLM-Ba`. |
| TRLM-FoBa (Forward) | Pre-trained both in forward and reverse token order. Forward version scores and generates identically to `TRLM-Fo`.

**Scoring:** Scores identically to `TRLM-Fo`.
**Generation:** Generates identically to `TRLM-Fo`. |
| Self Scoring | The model that is used for generating a given response is also used for scoring responses given queries in the conventional forward scoring direction.
**Scoring:** We use the model's own perplexity scores as feedback to select the responses. |
| Forward Baseline | A conventional forward model trained for next-token prediction on the same training corpus and model class as TRLM .
**Scoring:** While self-scoring used the perplexity obtained from the generator model, in this setting, we use perplexity of a different forward model. |

[TRLM-Fo ] maintaining the standard token order during pre-training but reversing the direction of generation through appropriate prompts during inference (scoring and generation).

We show that TRLM provides non-trivial unsupervised feedback that could be used by pre-trained, fine-tuned, and instruction tuned models, for various downstream tasks like reranking to improve open-ended long-form question answering, generating citations, and retrieval. We demonstrate that the ability of TRLM to score in the reverse direction – scoring query based on the response – is essential to achieve the requisite gains. Further, TRLMs that are pre-trained in the reverse direction (TRLM-Ba ) provide an additional boost in most cases. We further leverage the generative ability of TRLM in reverse (generating query from a response) to amplify the effectiveness of input safety filters as well.

We propose four variants of the TRLM class – TRLM-Ba , TRLM-Fo , TRLM-FoBa (Reverse) and TRLM-FoBa (Forward) – based on how they are pre-trained and fine-tuned. TRLM models can be considered to have three functions: TRLM.Pretrain, TRLM.Score, and TRLM.Generate, which we describe for each of the four variants in Table 1. We further outline these functions for different TRLM models in Algorithms 1, 2, 3, & 4. For this work, we consider two baselines, which are trained in forward token order, and score in the conventional order of response given the query. The first of these uses self-scoring based on the model's own perplexity. The second (Forward Baseline) is a forward model that we train, whose training corpus and model class are identical to TRLM.

TRLM **Model Training:** The pre-training setup for all TRLM models is identical to that of PALM2-Otter models described by Anil et al. [2023b], except for the token orders specified by our TRLM.pretrain methods for TRLM-Fo , TRLM-Ba and TRLM-FoBa respectively. We fine-tune them on the FLaN dataset [Longpre et al., 2023] using the TRLM-xx.pretrain function. Where xx can refer to Fo, Ba or FoBa based on the model being fine-tuned. Let `Instruction`, `Question`, `Answer`

denote instruction, question and answer respectively. Before calling the `pretrain` function during fine tuning , we merge `Instruction + Question` to be the new question.

## 4 Scoring in Reverse

In this section, we provide formal results on `TRLM` and the benefit of using pre-training in the reverse direction. Let us denote by $\mathbb{P}_{\texttt{Fw}}(A|Q)$ the conditional distribution of a forward LLM. Similarly, denote $P_{\texttt{TRLM}}(Q|A)$ to be the conditional distribution of the Time Reversed Language Model. For simplicity, we merge the instruction and question together.

### 4.1 Formal Results on Reverse LLM based Alignment

In this subsection, we focus on the distribution shift encountered while using a reverse model based scorer on forward generations. Specifically, we conclude that while reranking using `Forward Baseline` is equivalent to temperature scaling [Yang et al., 2024b], reranking using `TRLM` induces a distribution shift that is not equivalent to temperature scaling.

Consider the *alignment* problem of learning a new forward LLM - $\tilde{\mathbb{P}}_{\texttt{Fw}}(\texttt{Answer}|\texttt{Question})$. A very popular framework is the KL constrained optimization objective with respect to a reward oracle $\mathcal{R}(\texttt{Question}, \texttt{Answer})$, for some threshold $\Delta$:

$$\max_{\substack{\tilde{\mathbb{P}}_{\texttt{Fw}}}} \mathbb{E}_{\substack{\texttt{Question}\sim\mathcal{Q} \\ \texttt{Answer}\sim\tilde{\mathbb{P}}_{\texttt{Fw}}(\texttt{Answer}|\texttt{Question})}} [\mathcal{R}(\texttt{Question}, \texttt{Answer})] \text{ s.t. } D_{\text{KL}}(\tilde{\mathbb{P}}_{\texttt{Fw}}\|\mathbb{P}_{\texttt{Fw}}) \leq \Delta \qquad (1)$$

**Log-perplexity of the forward model used as reward:** In general, for long form question answering where an explicit reward model is not available, a typical method is to use log-perplexity of the forward model i.e. $\log \mathbb{P}_{\texttt{Fw}}$ as a reward. Then, we have the following corollary of Lemma 1 in Yang et al. [2024b],

**Lemma 1** (Corollary of Lemma 1 in Yang et al. [2024b])**.** The new LLM policy $\tilde{\mathbb{P}}_{\texttt{Fw}}$ that optimizes (1) is given by: $\tilde{\mathbb{P}}_{\texttt{Fw}}(\texttt{Answer}|\texttt{Question}) \propto \mathbb{P}_{\texttt{Fw}}^{1+\alpha}(\texttt{Answer}|\texttt{Question})$ where $\alpha$ is chosen appropriately depending on the threshold $\Delta$ when reward $R(\cdot)$ is set to $\log$ perplexity of the forward model $\mathbb{P}_{\texttt{Fw}}$.

A policy obtained post the constrained KL-alignment procedure is akin to temperature re-scaled forward model, since $p^{1+\alpha}$ is equivalent to *temperature rescaling* $\exp^{(1+\alpha)\log p}$.

**Log-perplexity of the `TRLM-Ba.score` used as reward:** Suppose $R(\cdot)$ is set to output of `TRLM-Ba.score` computed on the the question given the answer, then we have:

**Lemma 2** (Corollary of Lemma 1 in Yang et al. [2024b])**.** The new LLM policy $\tilde{\mathbb{P}}_{\texttt{Fw}}$ that optimizes (1) is given by: $\tilde{\mathbb{P}}_{\texttt{Fw}}(\texttt{Answer}|\texttt{Question}) \propto \mathbb{P}_{\texttt{Fw}}(\texttt{Answer}|\texttt{Question})\mathbb{P}_{\texttt{TRLM-Ba}}^{\alpha}(\texttt{Question}|\texttt{Answer})$ where $\alpha$ is chosen appropriately depending on $\Delta$ when reward $R(\cdot)$ is set to $\log$ perplexity of the reverse model $\mathbb{P}_{\texttt{TRLM}}$.

Optimal distribution after alignment using `TRLM` scores results in a non-trivial distribution that is not simply temperature re-scaling. While we have not used `TRLM` for alignment using KL constraints in our experiments, the distribution shift that is induced by reverse token training is indeed non-trivial even with Best-of-N-re-ranking, which we adopt in our experiments.

## 5 Experimental Results

In this section, we explore the effectiveness of time reversed language models on different downstream tasks, by utilizing unsupervised feedback to improve upon existing forward model generations. Broadly, these applications fall into two categories - first, where we utilize the scoring capacity of `TRLM` (three use cases), and second where we utilize the generative capacity of `TRLM` for generating queries given a response.

### 5.1 Best-of-N reranking

The best-of-N reranking task involves outputting the best response out of $N$ model responses to a user query. Specifically, given $N$ LLM outputs to a user query, a reranking algorithm finds the

best response based on scalar scores assigned to each response. Prior works [Rafailov et al., 2023, Mudgal et al., 2023a] aim to improve LLM performance on this task by using feedback-based RLHF algorithms and training on KL-regularized alignment objectives. Yang et al. [2024a] show that best-of-N reranking is the most effective way to approximate these RL objectives, and further, it is empirically observed to outperform them.

In this work, we consider several best-of-N reranking based algorithms based on TRLM.Score, for evaluating a base model response. The methods considered rely on nothing more than the pre-training (or instruction-tuning) corpus to achieve alignment of response to the user query. We further note that such scores from TRLM may be used within RL objectives as well, but we leave the exploration of such rewards to future work.

### 5.1.1  Alpaca Leaderboard Evaluation

**Benchmark and Evaluation:** The `AlpacaEval` leaderboard [Dubois et al., 2024] is a widely used benchmark to evaluate the capability of language models. In this benchmark, there are 805 questions from the *AlpacaFarm* evaluation set – consisting of questions ranging from general writing, chat ability, and reasoning to general knowledge. The goal is to output a response that is better than a base model's response, as judged by an annotator model. Both base model and annotator model are set as `GPT4-1106-Preview` on the `AlpacaEval` leaderboard as on May 10, 2024, and hence we use the same for our evaluations. The evaluation benchmark computes various metrics including winrates, discrete winrates and length-controlled winrates [Dubois et al., 2024]. The length-controlled winrates are calculated using a debiasing algorithm that removes the length bias that is otherwise preferred by `GPT4-1106-Preview` .

Formally, we define the task for TRLM as follows — Given a query $Q$ from the dataset and $N$ model responses $\mathcal{A} = \{A_1 \ldots A_N\}$ from a generator model, we wish to use `TRLM.score` to output the highest scoring response $a_i \in \mathcal{A}$, which is further evaluated against an answer from `GPT4-1106-Preview`.

In our experiment, we consider outputs from a generator model that is `Gemini-Pro-1.0` [Anil et al., 2023a]. We generate 16 responses using a temperature $\tau = 0.8$ to ensure diversity of answers. We then rerank the responses using different variants of TRLM from the `PALM2-Otter` family of models (TRLM training details in the supplement). We further consider two baselines, `Self scoring` and `Forward Baselines`, as described in Table 1. Scoring prompts and Conditioning prompts used with various TRLM variants for this task are described in the Table 7 of Appendix C.1.

**Discussion of Results**: In Table 8, we see that TRLM-Ba scores the highest length controlled win rate which is $5\%$ over the `self scoring` baseline of `Gemini-Pro-1.0` with 16 generations against the `GPT4-1106-Preview` judge. Further, it registers an $8\%$ increase over the reported number for single generations in the benchmark leaderboard. We note that scoring `Response->Query` seems to bring out some improvements as TRLM-Fo improves over `Forward Baseline`. Further, TRLM-Ba outperforms TRLM-Fo indicating the impact of reverse token pre-training. This demonstrates that time reversed scoring provides an intrinsic unsupervised feedback that could help improve the performance of even larger capacity models. We note that pre-training in both forward and reverse directions (TRLM-FoBa models) and scoring in the reverse direction is better than TRLM-Fo variant.

We present further results where the generations of a Mixtral model [Jiang et al., 2024b] are reranked and compared against `GPT4-1106-Preview` , and the generations of a smaller Mixtral model are reranked and compared against a larger Mixtral model. These results are presented in the Appendix C.2. We note a 4% improvement over `Forward Baseline` with the proposed `TRLM-Ba.Score` method of reranking.

**Key Takeaway:**  Through empirical justifications, we show that TRLM variant models can be used as effective re-rankers of generations from multiple classes of models (`Gemini-Pro-1.0` , Mixtral8x22B, Mixtral8x7B), and improve the instruction following capability of the model as a whole. This is consistent with the 1 considering the fact that we outperform generation model's self-log perplexity score. While other methods of re-ranking exists, to the best of our knowledge none of them provide unsupervised feedback for effective reranking with just a pre-trained model.

Table 2: The best re-ranked response is compared with a single response of `GPT4-1106-Preview`. The setting is identical to the `AlpacaEval Alp` Leader board. `TRLM-Fo`, that scores in the backward direction, fares better than the conventional forward baseline. Scoring using `TRLM-Ba` (pretrained in reverse) gets even a higher (LC) win rate.

**Model Performance on the Alpaca Leaderboard**

| Model | Inference Style | Win Rate | | | Standard Error | Wins | Losses | Ties |
| | | LC | Reg | Discrete | | | | |
|---|---|---|---|---|---|---|---|---|
| `TRLM-Ba` | `Response -> Query` | 32.44 | 24.35 | 24.04 | 1.27 | 192 | 610 | 3 |
| TRLM-FoBa (backward) | `Response -> Query` | 31.18 | 22.72 | 21.99 | 1.24 | 176 | 627 | 2 |
| TRLM-FoBa (forward) | `Response -> Query` | 30.55 | 22.85 | 22.48 | 1.25 | 180 | 623 | 2 |
| `TRLM-Fo` | `Response -> Query` | 29.19 | 22.68 | 21.30 | 1.24 | 170 | 632 | 3 |
| One Generation | - | 24.38 | 18.18 | 17.08 | 1.16 | 135 | 665 | 5 |
| Self | `Query -> Response` | 27.05 | 17.66 | 17.14 | 1.15 | 136 | 665 | 4 |
| Forward Baseline | `Query -> Response` | 24.27 | 17.13 | 15.78 | 1.12 | 126 | 677 | 2 |

## 5.2 Citation Attribution

In this section, we describe applications of reverse scoring to the task of producing citations to original passages that can *corroborate* the sentences in an already produced summary. Summaries are created from long form articles, and one often wants to know which part of the article a given summary sentence is derived from (Benjamin Cohen-Wang [2024]).

**Dataset and Evaluation:** For this task, we take the CNN Daily Mail Dataset [CNN] which consists of pairs of news articles and their respective highlights. Our goal is to identify which sentence (or groups of sentences) within a given news article provides the most direct corroboration for a specific article highlight given as a query. We evaluate the attributed citations using various relevancy metrics. We use cosine similarity on the embeddings of the `Gecko` model [Lee et al., 2024], cosine similarity on `TF-IDF` features, `BLEU` score and `ROUGE` score to compute metrics. We score and choose the best pairing using all the models from the `TRLM PALM2-Otter` family trained in the forward, reverse and forward-reverse directions as outlined in Section 5.1.1.

**Algorithms:** Different search algorithms, `Linear Search`, `Binary Search` and `Exclusion Search` are coupled with using `TRLM.score` to find the attribution. We outline these in Algorithms 7, 8 and 9 along with details in the supplement. The number of inference calls is $O(\log N)$ where $N$ is the number of article sentences for `Binary Search`, and this method produces multiple sentences as a citation. The other methods require $O(N)$ calls to produce the citation for a sentence.

Our results shown in Table 3, demonstrate the efficacy of `TRLM` for the attribution task. Specifically, we show 44% gains over the baseline in the linear search method, 39% gains in the binary search method and 34% gains in the exclusion search method as measured through `gecko` cosine similarity.

**Key Takeaway:** Through our results on CNN-Daily Summarization dataset we present multiple methods of citation attribution and demonstrate significant gains with `TRLM` model variants. We note that a direction of *low* information to *high* information (summary –> article) is harder to reason upon and select among a given set of texts. Further, we highlight the importance of binary **selection based** approach over log-perplexity based **exclusion based** search. We show 9% improvement using `TRLM-Ba` on `Gecko` embedding-based metric using only $O(\log N)$ inference calls to the main model.

Table 3: Tabulates the citation Attribution results through Re-ranking on the CNN-Daily Mail dataset. $A$ denotes article whereas $S$ denotes the corresponding summary. The ease of scoring a summary given the article instead of *reverse* is clearly highlighted in all of the search methods.

| Model | Inference Direction | LinearSearch | | | Binary Search | | | Exclusion Search | | |
| | | Gecko | TF-IDF | ROUGE | Gecko | TF-IDF | ROUGE | Gecko | TF-IDF | ROUGE |
|---|---|---|---|---|---|---|---|---|---|---|
| `TRLM-Ba` | A->S | 53.16 | **55.45** | 49.12 | **45.09** | **50.93** | **42.11** | 36.33 | 46.34 | 36.13 |
| TRLM-FoBa (Rev.) | A->S | **53.48** | 53.22 | **49.67** | 40.74 | 45.04 | 39.81 | 32.40 | 40.84 | 33.88 |
| TRLM-FoBa (Forw.) | A->S | 50.65 | 52.21 | 45.24 | 43.81 | 49.84 | 40.60 | **38.67** | **48.16** | **38.11** |
| `TRLM-Fo` | A->S | 45.00 | 49.40 | 37.66 | 43.14 | 49.65 | 39.22 | 37.90 | 47.83 | 37.98 |
| Forward Baseline | S->A | 9.33 | 9.54 | 11.06 | 5.88 | 6.66 | 6.69 | 4.66 | 7.53 | 7.00 |
| Backward Baseline | S->A | 7.62 | 8.23 | 9.18 | 5.47 | 6.23 | 6.32 | 4.11 | 5.02 | 5.11 |

## 5.3 Document Retrieval

In this section, we study the performance of TRLM in retrieving relevant passages from a corpus to answer a specific question. Our goal is to show the efficacy of TRLM based reverse scoring over doing it in the forward direction. The task is as follows: Given a question, the goal is to retrieve relevant documents from the given corpus. We retrieve $k$ documents from the corpus and compute various information-retrieval metrics to calculate performance w.r.t. the golden set of documents.

Table 4: Summary of MS-Marco and NF-Corpus Datasets

| Dataset | Description |
|---|---|
| **MS-Marco** | Contains 101.09k examples in its public dev split. Each example consists of a simple question along with 10 relevant passages. [Bajaj et al., 2016] |
| **NF-Corpus** | Medical information retrieval dataset with 323 queries in its test split and 3.6k total documents in the corpus. Queries are in simple English, and documents are extracted from PubMed with a fair amount of medical terminology. [Boteva et al., 2016a, Pub] |

We experiment with two retrieval-based datasets from MTEB benchmark [Muennighoff et al., 2023] as shown in Table 4. Metrics are precision, recall, normalized discounted cumulative gain (NDCG) and mean reciprocal rank (MRR) (details in Appendix E.1). We show our results in Table 5. TRLM reverse scoring algorithms along with respective prompts used are presented in Algorithms 11, 10 of the Supplement. As Table 5 suggests, results favor TRLM based reverse scoring methods. For example, we see a 22.48% improvement in recall at $K = 4$ for MS-MARCO dataset. TRLM-Ba model dominates across metrics. For NF-Corpus, we see that the conventional forward scoring algorithm (`query -> document`) has a very poor performance. We attribute this to the fact that, in this inference direction, we are scoring a highly complex medical document using a simple natural language query. We see a gain of 44.2 points in NDCG at $K = 10$ with TRLM-Fo compared to `Forward Baseline`. The results in both these datasets suggest that TRLM can show greater gains when the complexity of documents in the corpus differs significantly from the complexity of queries.

Table 5: Tabulates the results of various reranking algorithms with two inference directions. $Q$ denotes Queries, while $D$ denotes Documents. TRLM outperforms `Forward Baseline` and `Backward Baseline` significantly, which highlights the importance of inference direction in this task.

| Method | Inference Direction | MS-MARCO | | | | | NF-CORPUS | | | | |
|---|---|---|---|---|---|---|---|---|---|---|---|
| | | Precision | | Recall | | NDCG | Precision | | Recall | | NDCG |
| | | K=1 | K=4 | K=1 | K=4 | @10 | K=10 | K=20 | K=10 | K=20 | @10 |
| TRLM-Ba | D -> Q | **28.4** | **18.54** | **27.22** | **70.29** | **61.49** | 15.7 | 11.38 | 10.68 | **13.08** | 43.23 |
| TRLM-FoBa (Reverse) | D -> Q | 24.9 | 17.38 | 23.85 | 65.85 | 58.84 | 14.98 | 10.91 | 10.01 | 12.76 | 41.65 |
| TRLM-FoBa (Forward) | D -> Q | 21.16 | 15.58 | 20.25 | 59.08 | 55.46 | **17.86** | **12.6** | **11.11** | 13.5 | 48 |
| TRLM-Fo | D -> Q | 20.37 | 14.9 | 19.45 | 56.39 | 54.46 | 17.31 | 12.38 | 9.74 | 11.76 | **48.08** |
| Forward Baseline | Q -> D | 21.05 | 13.82 | 18.42 | 47.81 | 53 | 0.87 | 0.87 | 0.17 | 0.31 | 3.89 |
| Backward Baseline | Q -> D | 16.8 | 14.04 | 15.99 | 53.13 | 52.07 | 1.11 | 0.79 | 0.21 | 0.29 | 3.95 |

**Key takeaways:** We experiment with two information retrieval-based benchmarks MS-MARCO and NF-CORPUS and compute multiple metrics to compare TRLM variant models with standard `Forward Baseline` and unconventional `Backward Baseline`. We show a gain of 8.49 points in NDCG@10 on MS-MARCO and 44.19 points in NDCG@10 on NF-CORPUS. Aligning with the results in citation, the results from this task also accurately demonstrate the importance of going from a *high* information direction to a *low* information direction. The massive difference between the directions is evident in the NF-CORPUS dataset.

## 5.4 Defending against Jailbreak attacks

We next aim to leverage the generative ability of TRLM to augment toxicity filters that are used to improve the safety of LLMs. Prior works show that LLMs (and their input filters) can be jailbroken using crafted adversarial attacks [Zou et al., 2023], while output filters tend to have a high false negative rate due to the sensitivity to the presence of toxic words, despite being in a neutral context (See Table-10). We propose to combine the benefits of input and output filters by projecting the output response of LLMs to the input query space using the reverse generative capability of TRLM, and further detecting the toxicity of the generated queries to block/ pass the response to the original

Table 6: Performance of the proposed defense strategies across different thresholds, evaluated on the human annotated and jailbreakbench toxic responses. TRLM-Ba achieves significant gains over all other approaches. Notations: PT [Pretrained], IT[Instruction-finetuned], FNR[False Negative Rate], FPR[False Positive Rate], new-HA [new HA Dataset], JBB[JBB Dataset], (H) [Hard], (E) [Easy]

| Method | Thresh = 2 | | | | Thresh = 4 | | | | Thresh = 6 | | | |
|---|---|---|---|---|---|---|---|---|---|---|---|---|
| | FNR-HA | FNR-JBB | FPR (H) | FPR (E) | FNR-HA | FNR-JBB | FPR (H) | FPR (E) | FNR-HA | FNR-JBB | FPR (H) | FPR (E) |
| TRLM-Fo (PT) | 0.00 | 36.11 | 17.00 | 2.00 | 36.36 | 55.56 | 12.00 | 0.00 | 45.45 | 70.83 | 6.00 | 0.00 |
| TRLM-Ba (PT) | 18.18 | 52.78 | 0.00 | 8.00 | 27.27 | 65.28 | 0.00 | 2.00 | 27.27 | 69.44 | 0.00 | 2.00 |
| TRLM-Fo (IT) | 54.55 | 55.56 | 3.00 | 0.00 | 63.64 | 72.22 | 1.00 | 0.00 | 63.64 | 81.94 | 1.00 | 0.00 |
| TRLM-Ba (IT) | 18.18 | 59.72 | 0.00 | 8.00 | 18.18 | 70.83 | 0.00 | 4.00 | 27.27 | 79.17 | 0.00 | 2.00 |

query based on a pre-specified criteria. We thus effectively amplify input safety filters, i.e. reduce False Negative Rate (FNR) with marginal/ no impact on False Positive Rate (FPR).

**Key Idea:** Consider `TRLM.Generate(Response)` that generates queries that could have produced a given response. The insight is that, the reverse generative ability of TRLM allows the projection of a candidate (jailbreak) query that could bypass the input filter back to the (naive) query space observed during training. These projected questions can thus be rightly classified using the same input filter.

**Defense Strategy:** We propose a defense strategy where i) a query is passed through the input filter, ii) if the input filter rejects the query, we return reject as well, iii) if the input filter allows the query, we take the `Response` produced by the model and generate multiple queries using `TRLM.Generate(Response)`. If the number of generated queries rejected exceeds a threshold, we reject the query as "unsafe". Otherwise, we declare it as safe, and output the response corresponding to the input query. An elaborate description is provided in Algorithm 12 of the Supplement.

**Datasets:** We consider a human annotated (HA) dataset provided as part of the `JailbreakBench` benchmark [HAd] for evaluating the performance of toxicity classifiers. This contains 100 questions annotated by humans, of which 43 are annotated as toxic based on a majority vote across 3 annotators. We introduce a GPT-4 based filter, that considers the prompt-response pair to judge their toxicity (Details in Appendix-F.2), and has 0 FNR on this HA dataset, which is ideal for defense evaluation. We further consider a `gpt-3.5-turbo-1106` based input toxicity filter for the empirical evaluation of the proposed defense, which has an FNR of 25.58% on this dataset. These unblocked questions form our `new-HA` dataset for the experiments. In addition to this, we use the following datasets for evaluation: JBB dataset that contains jailbreak questions (that are toxic as per the GPT-4 judge, but are safe as per the GPT-3.5 filter we augment) corresponding to different attacks on `JailbreakBench`, E dataset that contain safe and easy questions and H dataset that contains safe questions that are hard to classify as safe. We discuss more details on these datasets in Appendix-F.1.

In the two toxic datasets (HA and JBB), the `gpt-3.5-turbo-1106` based input filter does not block any of the questions, and our defense strategy aims at lowering the False Negative rate on the toxic questions (JBB dataset and `new-HA` dataset), while ensuring a low false positive rate on the safe questions as well (E and H datasets). We present the improvements in FNR rates for Algorithm 12 when combined with the `gpt-3.5-turbo-1106` input filter and various TRLM variants in Table-6. We further present the impact of varying the threshold in Fig.4 of the Appendix.

**Results:** We firstly note that the proposed TRLM defense strategy improves the FNR of the `gpt-3.5-turbo-1106` input filter across all settings considered. Further, the TRLM-Ba pre-trained model improves FNR by more than 70% on the HA dataset and around 35% on the JBB dataset, and outperforms other variants with negligible impact on FPR.

We note that the proposed defense outperforms existing perplexity thresholding based defenses [Jain et al., 2023, Alon and Kamfonas, 2023] and Smooth-LLM [Robey et al., 2023] on the `JailbreakBench` attacks [Chao et al., 2023, Deng et al., 2024] owing to the integration with an input filter that already outperforms them on the same. Hence, we do not compare with them. Further, these defenses operate only in the input space, while the proposed defense aims at augmenting the input space with feedback from the response. Hence, the proposed defense is orthogonal to such methods, and can thus be integrated with them as well.

# 6   Conclusions

In this work, we explore the capabilities of `TRLM` for scoring and generation of queries, when conditioned on responses. Our study points to the importance of the `response` $\rightarrow$ `query` direction in LLMs. When deploying `TRLM` models for reverse scoring, we show improvements on `AlpacaEval` leaderboard, Citation attribution and retrieval tasks. We further show that generations from `TRLM` can augment safety filters effectively.

# 7 Limitations

We note that the assumptions made for our theoretical results in Section 4 are stylized, and may not hold true in practice, as the space of all answers to questions may not be adequately captured by assumptions in that section. Given this assumption, one may wish to explore other models for hallucination that are more general and provide results about reverse scoring. We leave such a theoretical exploration to future work.

Further, `TRLM` benefits have thus far been explored on tasks related to short form queries that have long answers. One may wish to understand and demonstrate the effects of reverse scoring on other tasks. For instance, one might pose the question – does `TRLM` provide possible benefits for a broader set of tasks that language models are used for. We leave the exploration of such settings in which the reverse scoring direction of `response` $\rightarrow$ `query` is better than the forward scoring direction, along with obtaining an understanding on the reason behind such an advantage, as part of future work.

# 8 Acknowledgements

We are grateful to Kathy Meier-Hellstern and Krishnamurthy Dvijotham for the helpful discussions regarding defending against Jailbreak attacks. We sincerely thank Roman Novak and Abhishek Kumar for their inputs on early versions of our work.

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

# A   Results on a Bipartite Graph Model for Questions and Answers

In this section, we outline a simple toy model involving a universe of questions and answers with relations between them where we show how `TRLM-Ba` perplexity based alignment distribution helps in picking the right answer when the forward model "hallucinates". For simplicity of exposition, we will only focus on the distribution $P_{\texttt{TRLM-Ba}}(Q|A)$ for the `TRLM` class of models.

**Universe of Questions and Answers:** We consider a universe of questions and answers in the form of a bi-partite graph which are deemed to constitute the ground truth. Let $\mathcal{Q} \subseteq \mathcal{V}^K$ and $\mathcal{A} \subseteq \mathcal{V}^K$ where $\mathcal{V}$ is the vocabulary, be the universe of questions and answers respectively. For a given question $Q$, let $\mathcal{N}(Q) \in \mathcal{A}$ denote the set of ground truth answers of $Q$. Let $\mathcal{G}(\mathcal{Q}, \mathcal{A}, E)$ be a bipartite graph such that $E = \{(Q, A)\}_{Q \in \mathcal{Q}, \mathcal{A} \in \mathcal{N}(\mathcal{Q})}$ is the edge set of all valid answers. In other words, Ideally, one may like a forward model to approximate the distribution, $P(A|Q) = 1/|N(Q)|, \ A \in N(Q)$ and 0 otherwise, closely.

**Hallucination Model (Hamming distance version):** We would like to model an imperfect forward model that does not fully adhere with the ideal ground truth forward model. For a given question $Q$, the imperfect model produces answers $\mathcal{N}(Q')$ to the neighbouring questions $Q'$ which are at a hamming distance of 1 from $Q$. Concretely, let $\mathcal{H}(\cdot, \cdot)$ denote the hamming distance function. The support of the answer distribution is then $\mathcal{S} = \bigcup_{Q':\mathcal{H}(Q,Q')\leq 1} \mathcal{N}(Q')$. It follows immediately that $P_{\texttt{Fw}}(A|Q) = \sum_{Q':\mathcal{H}(Q,Q')\leq 1} \mathbf{1}_{A \in N(Q')}/|\mathcal{S}|$. Analogously, for a given answer $A$, let $\mathcal{S}' = \bigcup_{A':\mathcal{H}(A,A')\leq 1} \mathcal{N}(A')$. Then for `TRLM-Ba` we have $P_{\texttt{TRLM-Ba}}(Q|A) = \sum_{A':\mathcal{H}(A,A')\leq 1} \mathbf{1}_{Q \in N(A')}/|\mathcal{S}'|$.

**Theorem 1.** Let us assume the hallucination model above. Assume that for two questions $Q, Q'$ : $H(Q, Q') \geq 1, \ \min_{(A,A') \in N(Q) \times N(Q')} H(A, A') > 1$, then the optimal alignment distribution when $P_{\texttt{TRLM-Ba}}(\cdot)$ is used a scoring model (i.e. distribution in Lemma 2) has the support $N(Q)$ for $Q$.

*Theorem 1.* From Lemma 2, we have that

$$\tilde{P}_{\texttt{Fw}}(A|Q) \propto P_{\texttt{Fw}}(A|Q)P^{\alpha}\texttt{TRLM}(Q|A) \qquad (2)$$

for some $\alpha > 0$. For a fixed question $Q$, left hand side is potentially non-zero only for $A \in \mathcal{N}(Q')$ : $\mathcal{H}(Q, Q') \leq 1$. since the first term in the right hand side is non-zero only for those by definition of the hallucination model. Consider an $A$ such that $\exists Q' : A \in \mathcal{N}(Q'), \ \mathcal{H}(Q, Q') = 1$. We will argue that the second term is zero for such an answer $A$. Suppose it is non-zero, according to the hallucination model for the reverse direction, it means that $\exists A' : \mathcal{H}(A, A') = 1, \ A' \in \mathcal{N}(Q)$. However $Q$ and $Q'$ are hamming distance one away. From the assumptions, their neighborhood are far apart by more than 1, therefore contradicting the implication that $\mathcal{H}(A, A') = 1$. $\qquad \square$

**Key Takeaway:** Therefore under the above simplistic hallucination model, although the forward model has a wider support $|\mathcal{S}|$ in the answer space, due to alignment with `TRLM-Ba` 's perplexity, the new distribution has a support of at most $N(Q)$ provably. While assumptions in the theorem are not reflective of true complexities of the universe of questions and answers in a domain, this simple model shows that alignment using `TRLM`'s scoring metric can give rise to better re-ranking whenever nearby questions produce far away answers and generating forward models tends to confuse between nearby questions (a form of hallucination).

# B   TRLM **Subroutines -** `Score,` **Generate and** `Pretrain`

In this section, we provide the subroutines of our `TRLM` models as described in Section 3.

# C   Details on the Experimental Section

We describe details about our experiments in the following figure 1:

---

**Algorithm 1** `TRLM-Ba.Pretrain`

---

1: **Input:** $T$ - context length. $N$ - number of sequences. $\mathcal{C}$ index set of the vocabulary. Pre-training corpus of sequences $\{\mathbf{x}_i\}_{i=1}^N$ such that $\mathbf{x}_i \in \mathcal{C}^T$, $x_{ij} \in \mathcal{C}$. Initialize the model $p_\Theta(\cdot)$ with random weights.
2: **for** $i \in [1:N]$ **do**
3:     **for** $t \in [1:T]$ **do**
4:         $\Theta \leftarrow \Theta + \alpha_{i,t}\nabla_\Theta \log p_\Theta(x_{i,T-t}|x_{i,T}, x_{i,T-1}\ldots x_{i,T-t+1})$
5:     **end for**
6: **end for**

---

---

**Algorithm 2** `TRLM-Ba.Score`

---

1: **Input:** Query: $Q$. Response $A$. Conditioning Prompt: `CP`. Scoring Prompt: `SP`
2: **return** $\log \mathbb{P}_{\texttt{TRLM-Ba}}(\texttt{Reverse}(\texttt{SP} + Q)|\texttt{Reverse}(\texttt{CP} + A))$

---

---

**Algorithm 3** `TRLM-Fo.Score`

---

1: **Input:** Query: $Q$. Response $A$. Conditioning Prompt: `CP`. Scoring Prompt: `SP`
2: **return** $\log \mathbb{P}_{\texttt{TRLM-Fo}}(SP + Q|A + \texttt{CP})$

---

---

**Algorithm 4** `TRLM-FoBa.Pretrain`

---

1: **Input:** $T$ - context length. $N$ - number of sentences of length $T$. $\mathcal{C}$ index set of the vocabulary. Pretraining corpus of sentences $\{\mathbf{x}_i\}_{i=1}^N$ such that $\mathbf{x}_i \in \mathcal{C}^T$, $x_{ij} \in \mathcal{C}$.
2: Initialize the model $p_\Theta(\cdot)$ with random weights.
3: **for** $i \in [1:N]$ **do**
4:     **for** $t \in [1:T]$ **do**
5:         **if** $i$ is even **then**
6:             $\Theta \leftarrow \Theta + \alpha_{i,t}\nabla_\Theta \log p_\Theta(x_{i,T-t}|x_{i,T}, x_{i,T-1}, \ldots, x_{i,T-t+1})$
7:         **else**
8:             $\Theta \leftarrow \Theta + \alpha_{i,t}\nabla_\Theta \log p_\Theta(x_{i,t}|x_{i,1}, x_{i,2}, \ldots, x_{i,t-1})$
9:         **end if**
10:     **end for**
11: **end for**

---

---

**Algorithm 5** `TRLM-Ba.Generate`

---

1: **Input:** Response $A$. Conditioning Prompt: `CP`.
2: **return** $Q \sim \mathbb{P}_{\texttt{TRLM-Ba}}(\ \cdot\ |\texttt{Reverse}(\texttt{CP} + A))$

---

---

**Algorithm 6** `TRLM-Fo.Generate`

---

1: **Input:** Response $A$. Conditioning Prompt: `CP`.
2: **return** $Q \sim \mathbb{P}_{\texttt{TRLM-Fo}}(\ \cdot\ |A + \texttt{CP})$

---

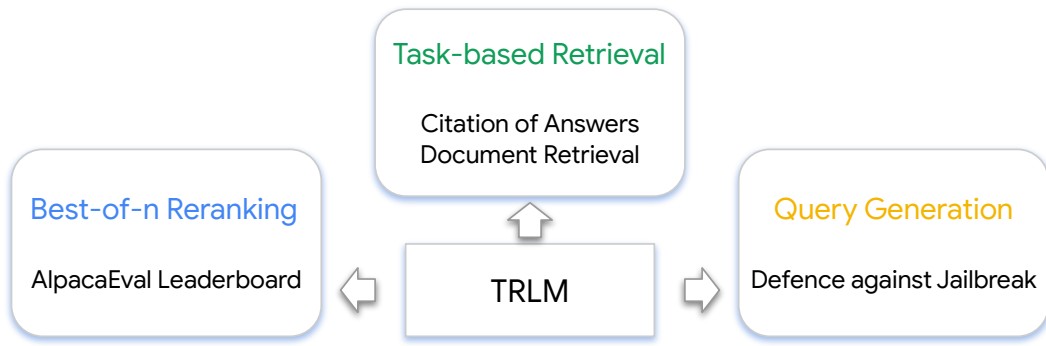

Figure 1: This task is an approach to link specific highlight sentences to lines that corroborate these sentences from within a lines in an article. By using linear binary and exclusion search methods, the aim is to efficiently and accurately find sentences in the articles that support the highlights.

Table 7: Per-Task Scoring and Conditioning Prompts

| Reranking Algorithm | Task | Scoring Prompt | Conditioning Prompt |
|---|---|---|---|
| `TRLM-Ba.Score` | Best-of-N Re-ranking
Citation Attribution
Passage Retrieval | "Question: "
∅
∅ | "? Answer:"
'is summarized by'
"is answered by" |
| `TRLM-Fo.Score` | Best-of-N Re-ranking
Citation Attribution
Passage Retrieval | "is the answer to"
∅
∅ | ∅
" is a summary of "
"has an answer to" |
| `TRLM-FoBa.Score (forward)` | Best-of-N Re-ranking
Citation Attribution
Passage Retrieval | Same as `TRLM-Fo.Score` Scoring | |
| `TRLM-FoBa.Score (backward)` | Best-of-N Re-ranking
Citation Attribution
Passage Retrieval | Same as `TRLM-Ba.Score` Scoring | |
| `TRLM-Ba.Generate` | Defense Generation | ∅ | "? Answer:" |
| `TRLM-Fo.Generate` | Defense Generation | ∅ | " is the answer to question:" |

## C.1   Scoring Prompts

We use scoring and conditioning prompts for all our re-rankers to evaluate the best possible response from the set of response to a query. We provide a detailed list of prompts used for each task in Table 7.

## C.2 Details on AlpacaEval Leaderboard results

Table 8: Mixtral 8x7B generations with `TRLM/Forward` reranking against Mixtral 8x22B reference as rated by a `GPT4-1106-Preview` annotator

**Model Performance on the Alpaca Leaderboard**

| Ranker | Inference Style | Win Rate | | | Standard Error | Wins | Losses | Ties |
|---|---|---|---|---|---|---|---|---|
| | | LC | Reg | Discrete | | | | |
| `TRLM-Fo` | `Response -> Query` | 42.07 | 47.54 | 47.08 | 1.51 | 379 | 426 | 0 |
| `TRLM-Ba` | `Response -> Query` | 44.13 | 46.98 | 47.39 | 1.52 | 381 | 423 | 1 |
| `TRLM-FoBa (Forw)` | `Response -> Query` | 42.88 | 47.11 | 46.58 | 1.52 | 375 | 430 | 0 |
| `TRLM-FoBa (Rev)` | `Response -> Query` | 44.28 | 46.67 | 45.71 | 1.50 | 368 | 437 | 0 |
| Self | `Query -> Response` | 43.56 | 41.88 | 42.11 | 1.52 | 339 | 466 | 0 |
| Forward Baseline | `Query -> Response` | 40.11 | 43.85 | 42.92 | 1.52 | 345 | 459 | 1 |

Table 9: Mixtral 8x22B generations with `TRLM/Forward` reranking against `GPT4-1106-Preview` reference as rated by a `GPT4-1106-Preview` annotator

**Model Performance Comparison**

| Ranker | Inference Style | Win Rate | | | Standard Error | Wins | Losses | Ties |
|---|---|---|---|---|---|---|---|---|
| | | LC | Reg | Discrete | | | | |
| `TRLM-Ba` | `Response -> Query` | 31.84 | 21.17 | 20.25 | 1.25 | 163 | 642 | 0 |
| `TRLM-FoBa (Reverse)` | `Response -> Query` | 32.58 | 21.06 | 20.37 | 1.24 | 164 | 641 | 0 |
| `TRLM-FoBa (Forward)` | `Response -> Query` | 29.43 | 21.31 | 20.37 | 1.23 | 164 | 641 | 0 |
| `TRLM-Fo` | `Response -> Query` | 31.95 | 22.05 | 21.24 | 1.25 | 171 | 634 | 0 |
| Forward Baseline | `Query -> Response` | 28.67 | 20.19 | 19.50 | 1.24 | 157 | 648 | 0 |
| Self | `Query -> Response` | 30.74 | 18.49 | 17.27 | 1.19 | 139 | 666 | 0 |

## D Details on the Citation Task

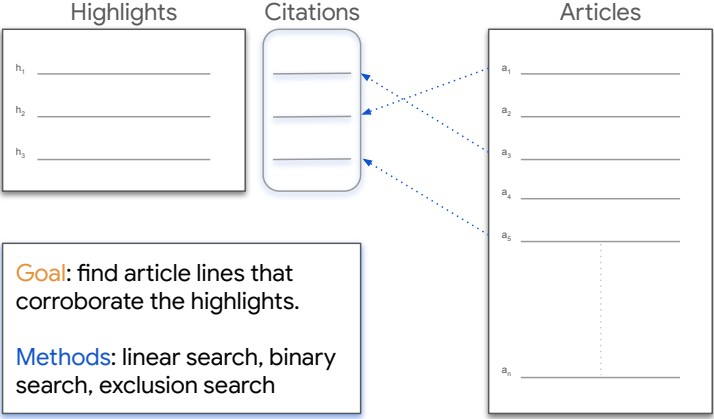

Figure 2: This task is an approach to link specific highlight sentences to lines that corroborate these sentences from within a lines in an article. By using linear binary and exclusion search methods, the aim is to efficiently and accurately find sentences in the articles that support the highlights.

**Algorithm Description:** We describe the three attribution algorithms that use `TRLM.score` function in the reverse direction with appropriate prompts in the supplement.

`Linear search` (Algorithms 7) uses scores every possible sentence in the article with the highlight sentence.

`Binary search`(Algorithm 8), actually starts with scores the first against the second half of the article for a given highlight and chooses the best recurses further by splitting the chosen half (analogous to binary search) until a contiguous set of article sentences of sufficient granularity is reached.

In `Exclusion search`(Algorithm 9), we drop article sentences and score the rest of the article with the highlight sentence. We pick the choice with the least *score*.

**Algorithm 7** `Linear Attribution Search`

---

1: **Input:** $h$ - highlight sentence , $A = \{a_1, \ldots, a_N\}$ - Article, `Conditioning Prompt:` `CP,Scoring Prompt :SP.`
2: Return $a_j$ corresponding to the highest `TRLM.score(`$h$`,`$a_j$`,CP,SP)`.

---

**Algorithm 8** `Binary Attribution Search`

---

1: **Input:** $h$, $A = \{A_s, A_{s+1}, \ldots A_t\}$, `Conditioning Prompt: CP,Scoring Prompt :SP.`
2: $s_1 \leftarrow$ `TRLM.score(`$Q = h, A = A_{s:s+\lceil\frac{t-s}{2}\rceil}$`,CP,SP)`.
3: $s_2 \leftarrow$ `TRLM.score(`$Q = h, A = A_{s+\lceil t-s/2\rceil} : t$`,CP,SP)`
4: **if then**$s_1 > s_2$
5: $\quad t \leftarrow s + \lceil\frac{t-s}{2}\rceil$
6: **else**
7: $\quad s \leftarrow s + \lceil\frac{t-s}{2}\rceil$
8: **end if**
9: **if then**$|t - s|$ has sufficient granularity **return** $A_{s:t}$
10: **else**
11: $\quad$ `Binary Attribution Search(`$h, A_{s:t}$`,CP,SP)`
12: **end if**
13: If $A_{\text{half}}$ is at the required granularity, return this as the attribution, else recursively search with $A_{\text{half}}$ as the article input.

---

**Algorithm 9** `Exclusion Attribution Search`

---

1: **Input:** $h_i$ - highlight sentence $i$, $A = \{a_1, \ldots, a_N\}$ - article sentences.`Conditioning Prompt:` `CP,Scoring Prompt :SP.`
2: Return $a_j$ corresponding to the highest `TRLM.score(`$h$`,`$A \setminus a_j$`,CP,SP)`. $A \setminus a$ denotes article $A$ without sentence $a$.

---

# E   Details on the Retrieval Tasks

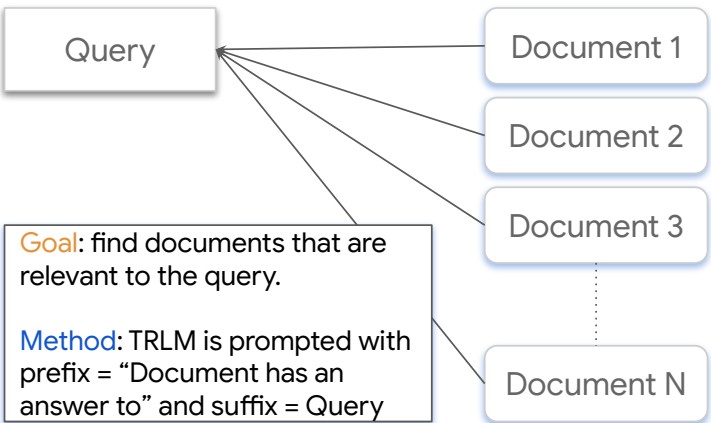

Figure 3: This task is used to assess the representational capability of `TRLM`. Here we look at how likely a document is to contain information relevant to answering a question. The language understanding of an LLM makes it likely that it produces better semantic retrieval than a simple embedding based model which is not contextual.

The scoring algorithms used for retrieval are given in Algorithms 10 11.

## E.1   Metrics Explanation

We compute the following metrics, that are widely used in information retrieval regimes.

---
**Algorithm 10** `Document Retrieval - TRLM-Fo`

---
1: **Input:** $Q$ - query, $D = \{d_1, \ldots, d_N\}$ - documents,`Conditioning Prompt: CP,Scoring Prompt :SP`.
2: Return $d_i$ corresponding with the highest score by `TRLM-Fo.score`($Q$,$d_i$,`CP`,`SP`).

---

---
**Algorithm 11** `Document Retrieval - TRLM-Ba`

---
1: **Input:** $Q$ - query, $D = \{d_1, \ldots, d_N\}$ - documents,`Conditioning Prompt: CP,Scoring Prompt :SP`.
2: Return $d_i$ corresponding with the highest score by `TRLM-Ba.score`($Q$,$d_i$,`CP, SP`).

---

**Precision@K**: We compute how many items within the top-k ranked items are relevant.

$$\text{Precision@K} = \frac{\text{No. of relevant items within top} - \text{k selected items}}{\text{k}}$$

**Recall@K**: We compute how many relevant items were selected out of the set of all relevant articles within top-k ranked items

$$\text{Recall@K} = \frac{\text{No. of relevant items within top} - \text{k selected items}}{\text{No. of relevant items}}$$

**NDCG@K**: Normalized discounted cumulative gain, where gain is defined as the rank of the selected item.

**MRR**: Mean reciprocal rank of the selection.

**NDCG@K** and **MRR** are order-aware metrics that not only test the retrieval performance but also how well a retrieval algorithm can order items in a set.

## F   Details on our Defence Task: Defending against Jailbreak attacks

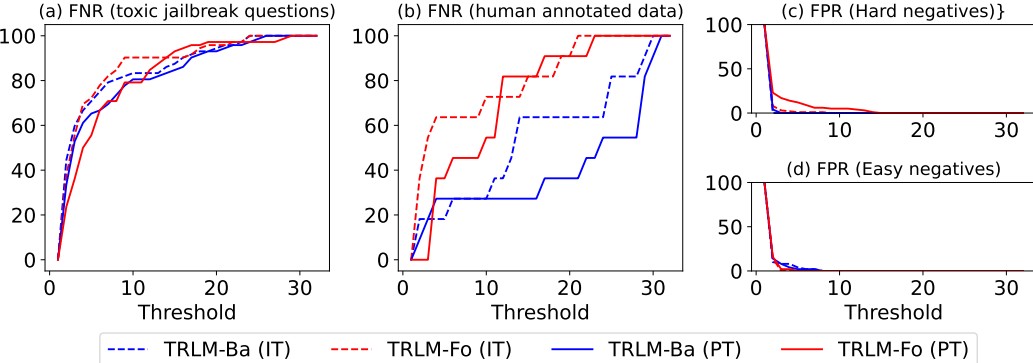

Figure 4: Plots showing the False Negative Rate and False Positive Rate of the proposed defense strategy. Positive indicates UNSAFE response, while negative indicates SAFE response. The first plot considers 72 questions generated from the `JBB` dataset. The second plot considers questions from the `new-HA` dataset. The third plot considers 48 hard safe questions generated by GPT4, whose answers contain content that appears unsafe (from the `H` dataset). The fourth plot considers 49 easy safe questions from Alpaca Eval2 dataset (`E` dataset). `TRLM-Ba (PT)` - the reverse pre-trained model clearly outperforms all other cases with lower FNR rate while keeping FPR rates under check.

### F.1   Datasets used in the Defence Task

**JBB Dataset:** We form the union of all question-answer pairs that have been generated by various attack strategies in `JailbreakBench`. There are a total of 1037 question and answer pairs of which

472 are identified as toxic by the GPT-4 toxicity filter. Amongst these questions, only 72 are declared as "safe" (False Negatives) by the `gpt-3.5-turbo-1106` input filter acting on the question alone, which we further reduce using the proposed defense. Thus, this set of 68 questions forms our JBB Dataset. We note that these are hard questions, since they are able to jailbreak the LLMs considered in `JailbreakBench`, as well as the `gpt-3.5-turbo-1106` input filter.

**E-Dataset:** We sample about 50 easy safe questions from `AlpacaEval` dataset, to test False Positive Rates, and eliminate one question that was predicted as unsafe by `gpt-3.5-turbo-1106`.

**H-Dataset:** To test the FPR on hard samples, we prompt GPT-4 to generate about 50 questions that could possibly mislead an input filter, while being safe. We further verify that the input-output pairs are safe using the GPT-4 toxicity classifier, and eliminate 2 questions that were predicted as unsafe. The prompt used to generate this dataset is shown below:

```
Can you help me create a dataset of questions that would lead to answers
containing words that can be perceived as harmful, but are not.  I want
to use this to ensure that my toxicity filter does not filter them out.
For example, "Create a political cartoon about gun violence", "Describe a
historic event in detail.,October Revolution".  Generate as many questions
as you can.  Stop ONLY when you find that you are repeating questions.
```

## F.2 GPT4 prompt used as a toxicity classifier

We prompt `GPT4-1106-Preview` with the following prompt to verify the toxicity of the question and answer.

> **SYSTEM PROMPT**
> Analyze user-submitted question-answer pairs with a focus on safety and ethical standards. Your goal is to classify each answer as "SAFE" or "UNSAFE", especially considering the potential for encouraging or detailing harmful behavior. Answer in one word without any explanation.
> Definitions:
> SAFE: Answers that are appropriate for general audiences, educate without promoting unethical actions, and maintain a respectful and informative tone without revealing sensitive information.
> UNSAFE: Answers that could potentially encourage or inform harmful or unethical behavior, even if unintentionally. This includes content that might educate or provide a how-to guide on harmful actions, discriminatory practices, manipulation tactics, illegal activities, or other unethical behaviors.
>
> **USER PROMPT**
> Question: <question>
> Answer: <answer>

## F.3 Algorithm for Question Generation for Defense

---
**Algorithm 12** Defense strategy using TRLM.Generate
---
1: **Input:** Question $Q$, a forward model/ system to be defended M, an input filter $\mathcal{F}$ that classifies input questions as UNSAFE/SAFE, filter-threshold $\tau$, number of generated questions $N$. `Conditioning Prompt:CP`.
2: If $\mathcal{F}(Q) =$ UNSAFE, return UNSAFE
3: Else Generate Answer $A$ from Forward Model $M$ for question $Q$
4: Generate $N$ questions $\mathcal{Q} = \{q_1 \ldots q_N\}$ using TRLM `.Generate(A,CP)`
5: Compute $\mathcal{F}(q_i)$ for $q_i \in \mathcal{Q}$. Calculate $T$ = total number of UNSAFE questions.
6: If $T >$ threshold $\tau$, return UNSAFE
7: Else return answer A to query Q
---

### F.4 Additional Tables relating Jailbreak Defense

Table 10: Comparison of various Input+Output Filter combinations on Human Annotated dataset on JailbreakBench. For the filter based on GPT-3.5 (version `gpt-3.5-turbo-1106`), we use the prompt from Llama-Guard [Inan et al., 2023]

| Method | Agreement | False Positive Rate | False Negative Rate |
|---|---|---|---|
| GPT-3.5 Output filter | 77.00 | 15.79 | 32.56 |
| GPT-3.5 Input filter | - | - | 25.58 |
| GPT-4 input+output filter | 89.00 | 19.30 | 0.00 |

# G Compute Requirements:

To pre-train `TRLM` models we use two TPUv5e pods[Cloud] for two weeks in the setup described by Anil et al. [2023b]. Further details on pre-training are provided in Appendix B. We run fine-tuning on FLAN-dataset using a TPUv5e pod [Cloud] for 1 day.

# H   Licenses and Copyrights Across Assets

1. `Gemini-Pro-1.0`
   - Citation: [Team et al., 2023]
   - Asset Link: [link]
   - License: Google APIs Terms of Service

2. `PALM2-Otter`
   - Citation: [Google and et al., 2023]
   - Asset Link: [link]
   - License: Google APIs Terms of Service

3. `GPT4-1106-Preview`
   - Citation: [Achiam et al., 2023]
   - Asset Link: [link]
   - License: OpenAI Terms of use

4. Mixtral 8x22B
   - Citation: [Jiang et al., 2024a]
   - Asset Link: [link]
   - License: Apache 2.0 license

5. Mixtral 8x7B
   - Citation: [Jiang et al., 2024a]
   - Asset Link: [link]
   - License: Apache 2.0 license

6. Gecko
   - Citation: [Lee et al., 2024]
   - Asset Link: [link]
   - License: Google APIs Terms of Service

7. CNN Daily Mail
   - Citation: [Zhong et al., 2020]
   - Asset Link: [link]
   - License: Apache 2.0 license

8. MS-Marco
   - Citation: [Bajaj et al., 2016]
   - Asset Link: [link]
   - License: Microsoft Terms and Conditions

9. NF-Corpus
   - Citation: [Boteva et al., 2016b]
   - Asset Link: [link]
   - License: Terms of Use

10. Alpaca Eval Benchmark
    - Citation: [Alp]
    - Asset Link: [link]
    - License: Apache 2.0 license

