# OpenReview forum: "Time-Reversal Provides Unsupervised Feedback to LLMs"
_NeurIPS.cc/2024/Conference — NeurIPS 2024 spotlight_

### Official Review · Reviewer_6ZRs · 2024-07-09

**Soundness:** 4
**Presentation:** 3
**Contribution:** 4
**Rating:** 8
**Confidence:** 3

**Summary:**

The paper proposes a class of LLMs called Time Reversed Language Models, which are simply pretrained on an unlabeled corpus in reverse order, and finetuned on instruction-tuned datasets accordingly. The model is used to provide feedback to LM generations in four different tasks. For general question answering, it serves to rerank N generated answers by scoring the probability of the question given the answers. For citation attribution, it scores a summarizing sentence given a cancidate text passage from the source article. For document retrieval, it scores the query given a retrieved candidate document. Finally, the model can be used to defend against jailbreak attacks by using the model to generate novel prompts given the response to the actual response. If the novel prompts exceed a threshold of rejection, the original prompt is rejected as well.

**Strengths:**

* The proposed methods are simple yet effective ideas of how to use a reverse LM for various tasks.
* While the idea to train a reverse LM is not novel per se, its applications are.
* The baselines are well chosen to demonstrate the benefits of the model. The claims are well-supported.
* On most tasks, the proposed method is very effective.
* The discussed methods are of great significance to the NeurIPS community.

**Weaknesses:**

* Even though the method is quite simple and easy to understand, I think the clarity of the paper could be better in some parts. For example, the purpose of the different parts/functions in Table 1 is not easy to understand without the context of the tasks that they are applied to. I think the paper could benefit from a "Problem Statement" section that introduces the different use cases from the tasks in Section 5 (e.g. reranking, retrieval, etc.) before presenting the model in Section 3.

**Questions:**

Suggestions:
* Section 6 is probably meant to be a subsection of Section 5.
* Results are presented as "our method performs better by x%", although what is meant is the absolute improvement in points on the scoring metric. E.g., in line 304, the improvement of your best variant over the Forward Baseline is not 44.19% but 44.19 points, which really is more than 1200% better than the Forward Baseline.
* I find Section 4 to be the least helpful part of the paper. As is acknowledged in the Limitations section, the assumptions are quite strong. I'd rather move it into the appendix in favor of using the additional space to enhance clarity as explained under "Weaknesses". However, I wouldn't object if the authors insist on keeping it the way it is.

**Limitations:**

Limitations are appropriately addressed in a separate "Limitations" section.

---

> ### Author Rebuttal · Authors · 2024-08-07
>
> We sincerely thank the reviewer for their valuable feedback on our work. It is indeed encouraging that the reviewer finds our work to be of great significance to the NeurIPS community and worthy of a strong accept.
>
>
> - **Clarity in some parts:** We thank the reviewer for this suggestion. We acknowledge that the explanation around Table 1 needs better clarity. We will add a Problem Statement section outlining all tasks we address in this work using the additional page in the camera ready version.
> - **Section-6 as a subsection of Section-5:** We agree with the reviewer, and we will update Section-6 to be a subsection of Section-5 in the final version.
> - **Improvement percentage:** We thank the reviewer catching this error. We will change the same to "improvement by 44.19 points" in the final camera ready version.
> - **Significance of Section-4:** We thank the reviewer for this suggestion. We would like to clarify that the theory presented in the paper is targeted at explaining the *intuition* behind the proposed approach, and is a merely intended to be a *proof of concept* that it works in *some stylized* settings. We thus acknowledge this in the Limitations section as well, as noted by the reviewer. We would like to emphasize that in the theory section, we also show that if the reverse score (P(Q|A)) is used as a reward function in an RLHF framework (which can be approximated by best of N reranking as shown in a recent work [1]), it leads to sampling Answer such that it is proportional to P(Question|Answer)*P(Answer|Question). This is significantly different from the conventional forward scoring, which we show to be merely a temperature scaling of the usual P(Answer|Question) distribution. This observation, while easy to derive, makes the motivation for reverse scoring very clear and conveys the key intuition that it considers both conditionals of P(A|Q) and P(Q|A) without the need for additional hyperparameter tuning during reranking. We will clarify this takeaway in the final version. We hope to be able to add the additional explanations related to experiments in the extra page that we are allowed in the camera ready version. In case of lack of space, we will certainly move parts of the theory as well to the Appendix.
>
> [1] J. Q. Yang, S. Salamatian, Z. Sun, A. T. Suresh, and A. Beirami. Asymptotics of language model alignment.
>
> We thank the reviewer again for their feedback and recognition of our work. We will be happy to address any further concerns as well.

---

> > ### Comment · Reviewer_6ZRs · 2024-08-13
> >
> > Thank you for your response. By making this already fine paper clearer, a lot of people will enjoy reading it.

---

### Official Review · Reviewer_4Q1B · 2024-07-12

**Soundness:** 3
**Presentation:** 3
**Contribution:** 3
**Rating:** 7
**Confidence:** 1

**Summary:**

This paper introduces Time Reversed Language Models (TRLMs), which operate in the response-to-query direction for scoring and generation. The key contribution is demonstrating that TRLMs can provide effective unsupervised feedback to improve language model performance on various tasks. Specifically, the authors show that TRLM-based reranking improves performance on the AlpacaEval leaderboard, citation attribution, and document retrieval tasks. They also demonstrate that TRLM's reverse generation capability can augment safety filters to reduce false negative rates in detecting toxic queries. The paper provides theoretical insights into why reverse scoring can be beneficial and empirically demonstrates the effectiveness of TRLMs across multiple applications, highlighting the importance of the response-to-query direction in language modeling.

**Strengths:**

1. Novel approach: The paper introduces Time Reversed Language Models (TRLMs), which operate in the response -> query direction for scoring and generation. This is an innovative approach to leveraging unsupervised feedback in language models.
2. Theoretical foundation: The authors provide formal results and a bipartite graph model to explain how TRLM-based alignment can help mitigate certain forms of hallucination.
3. Comprehensive experiments: The paper demonstrates the effectiveness of TRLMs across multiple tasks:
   - Best-of-N reranking on the AlpacaEval leaderboard
   - Citation attribution
   - Document retrieval
   - Defending against jailbreak attacks
4. Significant improvements: TRLMs show notable gains over baselines across tasks, e.g.:
   - 5% improvement in length-controlled win rates on AlpacaEval
   - 44% improvement in citation attribution accuracy
   - 44% improvement in NDCG@10 for medical information retrieval
5. Versatility: The authors show TRLMs can be used for both scoring and generation tasks, demonstrating their flexibility.
6. Safety applications: The paper explores using TRLMs to augment safety filters against toxic queries, showing potential for improving AI safety.
7. Broad applicability: The techniques are shown to work across different model families (e.g., PALM2-Otter, Gemini-Pro, Mixtral), suggesting wide applicability.

**Weaknesses:**

1. Assumptions: The authors acknowledge that the assumptions made for their theoretical results in Section 4 are stylized and may not hold true in practice. The hallucination model used is quite simplistic.
2. Need for more extensive testing of the defense strategy: While the proposed defense strategy for amplifying safety filters shows promise, the authors note it needs to be tested on a larger and more diverse set of questions to determine its broader applicability.
3. Computational costs: The paper does not thoroughly discuss the computational overhead of using TRLM for scoring and reranking, which could be significant for large-scale applications.

**Questions:**

1. The paper focuses mainly on English language tasks. How well might TRLM generalize to other languages or multilingual settings?
2. The paper shows TRLM can help with safety filtering, but are there any potential risks or downsides to using this approach for content moderation?
3. How sensitive are the TRLM results to the specific prompts used for scoring and generation? Did the authors explore prompt engineering to optimize performance?

**Limitations:**

Two suggestions for improvements:
1. More discussion of potential biases or fairness concerns in the TRLM approach would be valuable.
2. They authors could elaborate on privacy implications of using reverse query generation for safety filtering.

---

> ### Author Rebuttal · Authors · 2024-08-07
>
> We sincerely thank the reviewer for their feedback. We are happy that they find our work to be novel and versatile, our theory valuable, our experiments comprehensive and our improvements significant. We clarify the concerns in this rebuttal.
>
>   - **[Limitations section] Disclaimer on assumptions in theory:**  We clarify that the theory presented in the paper is targeted at explaining the *intuition* behind the proposed approach, and is intended to be a *proof of concept* that it works in *some stylized* settings. We thus include a disclaimer in the Limitations section to caution the reader about this. We will rephrase this is in the final version and clearly state that the method is *not* theoretically grounded, and requires empirical justification, which we present in Sections 5 and 6.
>   - **[Broader Impact section] Disclaimer on defense evaluations:**  We clarify that we indeed report results on the popular Jailbreakbench benchmark, that contains a diverse set of questions with toxic answers from several models families such as Llama, Vicuna and GPT. However, since the defense relates to the sensitive application of safety, we include a disclaimer to caution the reader to verify it on domain-specific applications before deploying the defense in practice. We will rephrase the same to clarify this in the paper.
>  - **Computational costs:** The cost for reranking is negligible compared to the overall inference cost which we explain below. Consider a setting where an LLM is deployed and served to end users. To answer the user's query, the LLM generates 16 responses simultaneously in a batch for $t$ decode steps ($t$ is the length of the response). Since the inference is autoregressive, this process introduces a factor of $t$ to the total inference time. Subsequently, TRLM is used for scoring all 16 responses in a single pass, while using nearly 10x lesser parameters. Assuming that compute scales linearly with the number of parameters, we can compare the computational time of:
>     - (a) Multiple response generation which is $t$, and
>     - (b) TRLM re-ranking which is $1/10$, since scoring is done in a single pass by a model of roughly 10x lower capacity
> Thus, (a) takes nearly $10\cdot t$ times more time than (b). Considering that $t$ (response length) is generally between 20 and 500, the additional cost of reranking is negligible.
> - **Generalization of TRLM to other languages:** We perform experiments to check this, and find that TRLM based reranking is indeed valuable in other languages as well. We present the experiment and results below.
>   - For a direct comparison with the Alpaca Eval results presented in the paper, we perform reranking for Alpaca eval in German. For a fair comparison, we use the same set of English questions and 16 English responses (from Gemini-Pro-1.0) that were used for presenting results in Table-2 of the paper. Only for reranking, we translate the question and responses to German using the Google Translate API and ensure that the TRLM models rerank German answers. Since the judge is designed and verified only for English answer evaluation, we use the corresponding English answer of the best response as input to the judge.
>    - We present the results in the below table. LCWR refers to Length controlled win rate while WR refers to normal win rate. We observe significant gains of 2.69% and 6.82% for German reranking using TRLM-Fo and TRLM-Ba respectively when compared to reranking all 16 responses using the standard forward direction (Forward Baseline). We note that these gains are similar to the reported gains. Thus, our TRLM-PaLM models are valuable for reranking in other languages as well.
>
> | Method | LC WR(German) | LC WR (English) | WR (German) | WR (English) |
> |---|---|---|---|---|
> | Forward Baseline P(A \| Q) | 24.5 | 24.27 | 17.30 | 17.13 |
> | TRLM-Fo P(Q \| A) | 27.19 | 29.19 | 20.22 | 22.68 |
> | TRLM-Ba P(Q \| A) | 31.32 | 32.44 | 24.12 | 24.35 |
>
>  - **Potential risks of safety filters:** The key risk involved in any defense is to incorrectly classify benign inputs as toxic, or a higher false positive rate (FPR). We show in Table-6 and Fig.4 that the proposed approach effectively amplifies input safety filters, i.e. reduces False Negative Rate (FNR) while not increasing FPR.
>  - **Sensitivity to prompts:** We use simple prompts in all cases to ensure that the observed results are not a result of excessive prompt engineering. We note that it may be possible to perform prompt engineering to improve scores in all cases. TRLM-Ba however does not require any prompts other than the "Question:" and "Answer:" tags, since Answer -> Question is its natural scoring direction. This is a key advantage of this model over TRLM-Fo. We further present results with two extreme cases of prompts for TRLM-Fo (Forward model prompted in reverse):
>
>   | Model | LC Win Rate | Win Rate |
> |---|---|---|
> | [Reported]: P(**[Q]**\|" **[A]** is the answer of the question: ") | 29.19 | 22.68 |
> | [Simple]: P(**[Q]**\|"Answer: **[A]** \n Question:") | 29.48 | 21.45 |
> | [Complex]: P(**[Q]**\|"Instruction: Generate a question that gives the following answer. \n Answer: **[A]**\n Question:") | 27.58 | 20.24 |
>
>    We note that predicting $P(Q|A)$ without any prompting (Simple) works best. This is very close to what we used for reporting, and thus represents a fair comparison with TRLM-Ba, which also does not use prompting.
> - **Potential biases:** We note that bias/ fairness issues do exist in LLMs, and apply to our work as well. The proposed approaches merely score the *quality* of the response, and thus do not present any *additional* bias/ fairness concerns to the best of our knowledge. We will include this as a caution to the reader.
> - **Privacy implications of defense:** Our defense works on LLM responses and not on user data. Since LLM responses already protect user privacy, the proposed defense does not pose additional threats.
>
> We will be happy to answer any further questions as well.

---

> > ### Comment · Reviewer_4Q1B · 2024-08-12
> > **Thanks for the rebuttal**
> >
> > I acknowledge the author's response and would like to increase the rating.

---

### Official Review · Reviewer_xuG3 · 2024-07-12

**Soundness:** 3
**Presentation:** 3
**Contribution:** 3
**Rating:** 7
**Confidence:** 3

**Summary:**

The paper explores the utilization of reverse/backward-trained causal LLM. This LLM can be used to score responses based on the probability of generating the queries given the scores (which can be combined with the probability of generating outputs from input). Then, they can be used for re-ranking. In general, they can be used to score sentence pairs - which can be used for citation attribution. The authors also consider the use of reverse LLM for toxicity filters. Given that standard output filters tend to be too aggressive in rejecting (high false negative), the authors found a way to improve the scores by generating the input from the output using the backward model, and then using the input filter for toxicity filtration.

Reasonable baselines are considered and compared (for example, using the plain forward model for generating the query from response in a format fashion using an appropriate prompt), or alternative scoring methods (like perplexity).

**Strengths:**

1. Application potentials of modern backward LLMs are not as well explored, so it's a relatively unique direction.
2. There are some interesting ideas for using backward LLMs, like toxicity filter besides standard ideas like re-ranking.
3. The paper provides some decent theoretical motivations for informing the scoring mechanism with a backward model.

**Weaknesses:**

1. The core idea is not particularly novel. There are also a few missed citations. [1,2,3] (There could be others I am missing. Would recommend doing a deeper literature search) similarly used backward-trained models (one of the references is dialoGPT, which falls under the causal language model family) and their scores for re-ranking.

2. It may be a bit hacky to circumnavigate the output filter by projecting the output into the input space. In principle, the output being toxic doesn't mean an appropriate input has to be. RIght now, the hacky approach still may working betters because the alternatives are limited,

[1] A diversity-promoting objective function for neural conversation models. - Li et al. NAACL 2016.

[2] DIALOGPT : Large-Scale Generative Pre-training for Conversational Response Generation - Zhang et al. ACL Demos 2020

[3] Generating Informative and Diverse Conversational Responses via Adversarial Information Maximization - Zhang et al. NeurIPS 2018

**Questions:**

Minor:

in Line 183, I assume P(A|Q) = 1/|N(Q)| should be the appropriate notation.not 1/N(Q).

Line 185: "the idea; ground truth"  - I believe you meant ideal.

**Limitations:**

Yes

---

> ### Author Rebuttal · Authors · 2024-08-07
>
> We sincerely thank the reviewer for their valuable feedback and relevant references. We are happy that the reviewer finds this direction to be relatively unique, the toxicity filter to be an interesting idea, and the theoretical motivation to be helpful. We hope to address the reviewer's concerns in this rebuttal.
>
>  - **Novelty of the core idea and related works:** We thank the reviewer for suggesting references that are relevant to our work. We will certainly add a discussion regarding these works in our revision. We briefly outline the discussion below:
>
>    - Li et al. [1] use mutual information between source and target for decoding. TRLM-Ba, trained in reverse token order, scores in reverse the forward generations and it results in outputs that satisfy a non-trivial objective for decoding. To see this, we argue in the theory section (Lemma 2), that doing RLHF using only the reverse score effectively samples answers proportional to $P(A|Q) P(Q|A) = \frac{P(A,Q)^2}{P(A) P(Q)}$. We note that this expression when summed over all answers and questions would be the $\chi^2$ divergence between dependent and independent distribution. This shows that forward generation’s contribution is the $P(A|Q)$ term while reverse scoring’s contribution is $P(Q|A)$ and it actually achieves the objective (in a qualitative $\chi^2$ sense instead of the KL sense) as in [1].
>    -  DialoGPT proposed by [2] pretrained a "backward" model to score sources given targets -- this is specific to conversational understanding. This is different from results on general instruction following benchmarks we present, which are significantly more general and open ended. For very large language models and on public, challenging benchmarks like AlpacaEval that test their instruction following capabilities, our work shows that full reverse token pre-training and reverse scoring gets the best set of results in re-ranking which is novel.
>
>     -  Zhang et al., [3] propose adversarial mutual information maximization, by using a "backward proposal network" during training of the model to maximize mutual information to improve the *informativeness* of generated responses. Further the proposed method involves GAN style training which could be unstable. We sidestep this issue by either prompting a forward model in the reverse direction (TRLM-Fo) to score or pre-training a model to learn in reverse that naturally scores in reverse (TRLM-Ba). We propose a method that can be used with any LLM and can be applied to any query.
>
>     - In summary, all previous works seem to motivate the need for better decoding based on scores in both directions. We show that reverse scoring alone, when used with forward generations, will achieve this naturally using a formal RLHF based argument and strong empirical results to back it.
>
>      - As the reviewer has noted, we explore several other novel applications such as citations, retrieval and defense strategy to amplify input filters. This demonstrates the generalization of TRLM across different tasks.
>
>  - **Projecting output responses to input space**: We would like to clarify that the projection of output into the input space is in fact a key ingredient of the proposed defense.
>     -  We claim that given a toxic output, there is a non-zero probability of generating a relevant question that is classified as toxic (although non-toxic questions may also be generated as the reviewer pointed out).  This probability is significantly higher than the probability of generating a toxic question given a safe response. We specifically rely on this property to design our defense, which also empirically holds from the results. Thus, the proposed method augments input filters effectively while maintaining a low False Positive Rate. Thus we think the projection of output responses to the input space is indeed a principled approach.
>
>    -  We would like to further clarify the intuition behind the proposed defense. Output filters get confounded by the presence of artifacts in a long response, while input filters can focus only on higher level reasoning of whether a question can elicit toxic content. This is evident from the significantly better performance of input filters empirically when compared to output filters in recent works (Table-1 in [4]).
>    - We also perform a small ablation to highlight the better reasoning capabilities of input filters by taking toxic questions and jailbreak responses from Jailbreakbench for Llama-2, and checking the accuracy of predicting this as toxic by input and output filters. We additionally encode the prompts and responses by inserting a space between each character to check the understanding of the filter. We note that in both cases, the LLM is the same, and the input and output filters differ only with a prompt/ response indication in the system prompt. It can be noted that the true positive rate drops by 51% in case of input filter, and by 88% in the output filter - highlighting the better reasoning of input filters.
>
> | | Input filter TPR (original) | Input filter TPR (encoded) | %misses (input filter) | Output filter TPR (original) | Output filter TPR (encoded) | %misses (output filter) |
> | ------------------------------------ | ---------------- | --------------- | --------------- | ----------------- | ---------------- | ---------------- |
> | Add one space between each character | 84 | 41 | 51 | 67 | 8 | 88 |
>
>    -  Our strategy thus allows the model to reason, where the projection of response to inputs acts as a chain-of-thought step while filtering output content.  Thus, when we project a toxic response back to the input space, we combine the benefits of input and output filters while overcoming their individual shortcomings.
>
>  - **Minor Points**: We will correct the notational issues and typos pointed out by the reviewer.
>
> [4] ShieldGemma: Generative AI Content Moderation Based on Gemma
>
>
> We will be happy to address any further concerns as well.

---

> > ### Comment · Reviewer_xuG3 · 2024-08-08
> > **Response**
> >
> > Thank you for the rebuttal. Based on the rebuttal, I increased the score to 7. I am overall convinced by your points about the application of the approach for toxicity filter as more than a hack, and so I take back weakness 2. Since you are sampling multiple queries and looking at the percentage of rejection, it should be relatively robust against stochasticity, and the assumption that the probability of the input query being negative is high is reasonable.
> >
> > Regarding the discussion of related works. I am not too keen on some of the phrasings. For example:
> >
> > > "DialoGPT proposed by [2] pretrained a "backward" model to score sources given targets -- **this is specific to conversational understanding. This is different from results on general instruction following benchmarks we present, which are significantly more general and open ended.** For very large language models and on public, challenging benchmarks like AlpacaEval that test their instruction following capabilities, our work shows that full reverse token pre-training and reverse scoring gets the best set of results in re-ranking which is novel."
> >
> > It's not clear to me that instruction following is necessarily more open-ended and general than conversational understanding. You can always set up a conversational context to follow instructions or vice versa. Many powerful LLMs are already set up with conversational style prompts in their final instruction-tuned models. I don't recall DialoGPT doing anything so "specific" that it applies exclusively to explicit conversational setups. While sure they didn't have as exhaustive as an exploration as yours specifically for backward scoring and the current datasets were not even available back them but the point was more about the novelty of the core method, not differences in experiments, datasets, or parameters (how large the language model is).
> >
> > Either way I am still giving a higher score, because the ideas are still not explored previously in more modern contexts and the paper has novel application ideas, and experiments with well-made baselines and variations.

---

> > > ### Author Response · Authors · 2024-08-13
> > > **Response to reviewer**
> > >
> > > We sincerely thank the reviewer for their feedback on our rebuttal and the increase in score. We will certainly incorporate the feedback from the reviewer in our camera ready version.

---

### Author Rebuttal · Authors · 2024-08-07

We sincerely thank the reviewers for their valuable feedback on our work, which has helped improve our submission. The reviewers have appreciated the novel application of reverse scoring and generation through our proposed TRLM family of models to various tasks like retrieval, citation and amplifying input safety filters. Some reviewers have also appreciated our theoretical results (although stylized to simpler models) to be insightful. We are grateful for their time and comments. We outline some salient points of our rebuttal:

  - **Whether re-ranking and scoring would generalize to other languages:** We present results by translating Gemini Pro 1.0 generations (used in the main paper evals) and queries from AlpacaEval to German and reranking using our TRLM models in German. We note from the results that win rates using reranking in German do not vary considerably when compared to reranking in English, and the gains of TRLM models are preserved (Please note that only scoring and reranking by TRLM has been done in translated German. Prompts/Evaluations by AlpacaEval judges are difficult to change into German in a manner that also preserves the correlations of the win rates with human preferences in the ChatBot arena. So we just use the corresponding reranked counterparts in English to score as usual.)

  - **Theoretical Properties of Reverse Scoring on Forward Generations** - We would like to emphasize Lemma 2 of the paper that shows that the post RLHF distribution, when reverse scores are used for feedback, effectively samples answers A for query Q with probability proportional to P(A|Q)*P(Q|A). In additional references that one of the reviewers has brought about, sampling using composite scores is desirable. We show that our TRLM framework can achieve it formally backed by empirical evidence.
  -  **Justification for projecting toxic responses to the question space:** We claim that given a toxic output, there is a non-zero probability of generating a relevant question that is classified as toxic (although non-toxic questions may also be generated as the reviewer pointed out).  This probability is significantly higher than the probability of generating a toxic question given a safe response. We specifically rely on this property to design our defense, which also empirically holds from the results. Thus, the proposed method augments input filters effectively while maintaining a low False Positive Rate. Thus we think the projection of output responses to the input space is indeed a principled approach. Further, output filters get confounded by the presence of artifacts in a long response, while input filters can focus only on higher level reasoning of whether a question can elicit toxic content. This is evident from the significantly better performance of input filters empirically when compared to output filters in recent works. We refer the reader to our response to xuG3 for empirical justification of the same.

We look forward to hearing back from the reviewers and more discussion in the next phase. We will be happy to clarify any further concerns as well.

---

### Decision · Program_Chairs · 2024-09-25

**Decision:**

Accept (spotlight)

**Comment:**

This paper explores the use of reversal language model for different four applications and tasks. Although the core idea of reversal language model is not new, authors novelly adapted it for several applications, resulting in a significant enhancement of performance. The paper is well presented and straightforward.

All reviewers are positive about this submission. Concerns and issues raised by reviewers were properly addressed during the discussions with authors. Authors also promise to incorporate this improvement in the camera-ready version.

I would love to recommend its acceptance.